# Preoperative Prognostic Index for Patients with Brain Metastases—A Population-Based Multi-Centre Study

**DOI:** 10.3390/cancers15123174

**Published:** 2023-06-13

**Authors:** Rebecca Rootwelt Winther, Eva Skovlund, Joakim Stray Andreassen, Lisa Arvidsson, Jonathan Halvardson, Ole Solheim, Jiri Bartek, Stein Kaasa, Marianne Jensen Hjermstad, Einar Osland Vik-Mo

**Affiliations:** 1European Palliative Care Research Centre (PRC), Department of Oncology, Oslo University Hospital, Institute of Clinical Medicine, University of Oslo, 4956 Oslo, Norway; 2Department of Public Health and Nursing, Norwegian University of Science and Technology, 7034 Trondheim, Norway; 3Department of Neurosurgery, St. Olavs University Hospital, 7030 Trondheim, Norway; 4Department of Neurosurgery, Karolinska University Hospital, 17164 Stockholm, Sweden; 5Department of Clinical Neuroscience, Karolinska Institutet, 17177 Stockholm, Sweden; 6Department of Neuromedicine and Movement Science, Norwegian University of Science and Technology, 7034 Trondheim, Norway; 7Institute of Clinical Medicine, University of Oslo, 0318 Oslo, Norway; 8Vilhelm Magnus Laboratory, Department of Neurosurgery, Oslo University Hospital, 0372 Oslo, Norway

**Keywords:** brain metastases, surgery, prognostication, survival, index

## Abstract

**Simple Summary:**

Metastases to the brain result in serious morbidity and mortality. Surgical resection is a treatment option; however, many patients die within a short time after surgery. It is therefore important to understand which patients may benefit from surgical resection. We have investigated prognostic factors for survival after surgery for brain metastases. We identified eight factors that significantly predicted survival in these patients and created a preoperative prognostic index to estimate survival and to guide clinical decision-making in evaluating surgery as a treatment for brain metastases. We tested our findings on patients who underwent the same treatment at two other large hospitals in Norway and Sweden, and our findings were valid. This is to our knowledge the first such index to support decision-making in this setting.

**Abstract:**

Background: Brain metastases (BM) are common in cancer patients and are associated with high morbidity and mortality. Surgery is an option, but the optimal selection of patients for surgery is challenging and controversial. Current prognostication tools are not ideal for preoperative prognostication. By using a reference population (derivation data set) and two external populations (validation data set) of patients who underwent surgery for BM, we aimed to create and validate a preoperative prognostic index. Methods: The derivation data set consists of 590 patients who underwent surgery for BM (2011–2018) at Oslo University Hospital. We identified variables associated with survival and created a preoperative prognostic index with four prognostic groups, which was validated on patients who underwent surgery for BM at Karolinska University Hospital and St. Olavs University Hospital during the same time period. To reduce over-fitting, we adjusted the index in accordance with our findings. Results: 438 patients were included in the validation data set. The preoperative prognostic index correctly divided patients into four true prognostic groups. The two prognostic groups with the poorest survival outcomes overlapped, and these were merged to create the adjusted preoperative prognostic index. Conclusion: We created a prognostic index for patients with BM that predicts overall survival preoperatively. This index might be valuable in supporting informed choice when considering surgery for BM.

## 1. Introduction

The diagnosis of brain metastases (BM) often occurs at a late stage of the cancer disease trajectory, and the prognosis is generally poor [1,2,3]. As many as 10–30% of cancer patients will develop BM, and the median overall survival (OS) is about five months [4]. However, survival varies widely according to primary diagnosis, age, presence and extent of extracranial disease and other tumor-specific biological factors [4,5,6]. Treatment usually aims at symptom control, but life prolongation is also possible, and both are commonly attempted. However, choosing the right treatment pathway for each patient is challenging, due to the heterogeneity of diseases and a lack of randomized controlled trials comparing all management options. Treatment options include stereotactic radiotherapy, whole brain radiotherapy, medical oncological treatments, such as traditional chemotherapy, immunotherapy and targeted therapies, and surgical resection [7,8,9,10]. A combination of modalities may be preferable [11,12,13,14,15,16]. In addition, best supportive care including corticosteroids, pain medication, anti-epileptics and antiemetics may be used for symptom control.

Patients with high functional status, stable extracranial disease and few intracerebral lesions may benefit from surgical resection of their BM [17]. Further, surgery may be necessary in cases of histopathological or molecular uncertainty regarding the brain tumor. Resection may also give rapid symptomatic relief and is usually preferred in BM that are too large for stereotactic radiosurgery [18,19]. Even so, about half of the patients operated on for newly diagnosed BM die within one year after surgery [20]. There is a lack of evidence to select those patients who are most likely to achieve a survival benefit from BM surgery. The updated ESMO Clinical Practice Guidelines for diagnosis, treatment and follow-up for patients with BM from solid tumors give general recommendations for when surgery is recommended, which include involving a multidisciplinary tumor board [21]. The biological diversity of BM is reflected in the complexity of the medical subspecialties involved. This poses a particular challenge in establishing well-functioning multidisciplinary tumor boards where all relevant specialties are represented. Most institutions have well-established meetings for organ-specific cancers. However, BM care is hampered by care allocated to the unifocal realms of neurosurgery and/or radiation oncology in isolation from each other and from medical oncologists who retain the depth of expertise for individual patients and their diseases and who increasingly offer CNS-active therapies [22]. While integrated multidisciplinary treatment is best evaluated in specialized meetings, this service is not available for most patients and caregivers. In fact, a recent study found that the use of resective surgery was offered to more patients after the establishment of a BM multidisciplinary tumor board [23], suggesting that surgery is underutilized in this patient group. Thus, there is an obvious need for data to support the evaluation of whether surgery could potentially benefit the patient.

Prognostication for patients with BM has a long history in medical science. In 2008 Sperduto et al. introduced the graded prognostic assessment (GPA) [24] that was an improvement of the prognostic classification recursive partitioning analysis (RPA) created by Gaspar et al. in 1997 [25]. After 2008, Sperduto et al. have further developed the GPA into the diagnosis-specific graded prognostic assessment (ds-GPA), where each primary cancer is assessed separately [5]. The RPA and the GPA are both based on data that is now very old. The rapid development of cancer treatment improves prognosis for patients, and these tools are no longer valid. Further, while providing highly useful information on which factors affect prognosis for patients with BM, the RPA, GPA and the ds-GPA are prognostic instruments generated for a broader oncological use. As such, these systems were not created specifically for selection to surgery. In addition, the newer ds-GPA requires molecular data and histopathological information that may not be available in up to one-third of patients prior to surgery for brain metastases [26]. In a previous study, we found that the GPA could not correctly predict survival and the ds-GPA could be calculated in only half of the patients referred to surgery for BM [3]. Therefore, selecting the right patient for BM surgery remains a substantial clinical challenge, and a new prognostic index created for this exact purpose is warranted.

The aim of this study was to improve prognostication for patients considered for BM surgery. In a previous study, we analyzed 590 patients who underwent surgery for BM at a geographically defined, single-provider referral center for neurosurgery in South-East Norway and identified several preoperative factors associated with prolonged survival [3]. In the current study, we sought to simplify the practical use of these prognostic factors and developed a preoperative prognostic index based on data from our previous study. Further, we validated the prognostic index against two populations from other geographically defined, single-provider institutions.

## 2. Materials and Methods

We created a preoperative prognostic index using a Cox regression survival analysis with preoperative prognostic factors based on a reference population (derivation data set). The derivation data set consists of 590 patients who underwent first-time craniotomy for BM from 2011 to 2018 at Oslo University Hospital (OUH). The median age was 63 years at the time of BM surgery (range 18–89). The most prevalent primary cancers were lung (33%), melanoma (16%), colon (9%) and breast cancer (9%). Fifty-one percent were female and 48% had comorbidities. Eighty-one percent had an ECOG status of two or better. Single BM was identified in 64% of the patients, while only 7% had more than four. Synchronous BM was found in 36%, while extracranial disease was considered stable in 29% and progressive in 18% of the patients. The preoperative factors used in the Cox regression analysis were based on a review of the literature and a round-table discussion among in-house oncologists and neurosurgeons at OUH. They included gender, age, primary cancer, performance status, presence of extracranial disease, location of BM, previous treatment and comorbidities. Only statistically significant factors were then dichotomized (yes/no) and used to create the index in a new regression analysis. Beta values were used to create the index scores. We pragmatically divided the index into four prognostic groups based on index scores.

To validate the preoperative prognostic index, we reviewed all patients who underwent first-time craniotomy for brain metastases at Karolinska University Hospital, Stockholm, Sweden, (KU) and at St. Olavs University Hospital, Norway, (StO) in the period 2011–2018 (validation data set). We used the preoperative prognostic index to score all patients in the validation data set and divide them into tentative prognostic groups. We created Kaplan–Meier survival plots to investigate whether the proposed prognostic groups were prognostic in real life. Digital patient records were evaluated locally at KU and StO, and each patient was scored in accordance with the preoperative prognostic index created at OUH.

After this validation, the preoperative prognostic index was adjusted in accordance with our findings to create an adjusted preoperative prognostic index.

### 2.1. Ethics

The study was approved by the data protection officer at OUH. Patient consent was waived due to the quality improvement focus of the study and the few patients still alive. All patients treated at OUH who were alive during the data collection were contacted and given the right to decline use of their data. The Regional Committee for Research Ethics (REC) South-East Norway and the Stockholm Regional Ethics Committee approved the study and transfer of data (REK number: 462536 and Dnr. 29017/1760-31 (sup-2020-02407) in Sweden)). Data were stored and analyzed in accordance with the GDPR.

### 2.2. Statistical Analyses

OS was analyzed using Kaplan–Meier plots and log-rank tests. Patients that were alive at the time of data extraction (OUH: 27.07.2020, KU: 08.02.21 StO: 01.01.22) were treated as censored. Hazard ratios were estimated by Cox’s proportional hazards model. Harrell’s C index was estimated to assess concordance. Frequencies were compared between groups using the Chi-square test. A *p*-value < 0.05 was considered statistically significant. We performed all statistical analyses in SPSS Statistics 28 (IBM Corp. Armonk, NY, USA).

## 3. Results

Based on the derivation data set described in the Methods section, we performed a Cox regression analysis with dichotomized variables that revealed eight statistically significant preoperative prognostic variables (Table 1). The variables associated with poorer OS were ECOG performance status above two, colorectal cancer, known extracranial metastases, progressive disease at the time of surgery or brain metastases as the first sign of cancer, age above 70 years, chemotherapy at any time prior to surgery and more than four brain metastases at the time of surgery. Breast cancer was associated with longer OS compared to other primary cancers. We used the Cox regression analysis to create the preoperative prognostic index with four prognostic groups: 0, 1, 2 and 3 (Table 1). The prognostic groups were pragmatically chosen based on scores. Patients in prognostic group 0 had the longest OS, and patients in prognostic group 3 had the poorest OS. OS for patients treated at OUH is illustrated in Figure 1 and Table 2.

From 2011 to 2018, 262 patients underwent first-time surgery for brain metastases at KU, and 180 patients underwent surgery at StO. Four patients were excluded due to missing survival data. Thus, 438 (261 + 177) patients were included in the validation of the preoperative prognostic index. Fifty-three (12%) patients were still alive at the last follow-up. Patients were divided into prognostic groups by summarizing the regression coefficients for each prognostic factor present preoperatively. There were statistically significant differences regarding the percentage of patients with age ≥ 70 years, ECOG > 2, breast cancer, progressive extracranial disease/synchronous BM and previous chemotherapy between OUH and the other two hospitals. However, there were no differences in the distribution of patients within each prognostic group between OUH and the two other hospitals. We describe general patient characteristics in Table 3.

The Kaplan–Meier plot shows significant differences in OS in the validation data set, which are consistent with the different prognostic groups found in the derivation data set except for prognostic groups 2 and 3. The median OS after surgery was ten months for all three hospitals (Table 4 and Figure 2).

We adjusted the index by merging prognostic groups 2 and 3, thus creating the adjusted preoperative prognostic index. This is demonstrated in Table 5 and Figure 3a,b.

## 4. Discussion

In this study, we have created an improved prognostic index for patients with brain metastases to estimate expected survival after surgery. The index was validated using two external patient populations from Sweden and Norway.

### 4.1. Patient Characteristics

In the population of patients used to validate the preoperative prognostic index, the percentage of patients within each prognostic group was similar to the derivation data set from OUH. This implies a comparable spread of risk evaluation for the acceptance of surgery for BM across the hospitals/regions. The median postoperative OS was ten months for all three hospitals, which is in concordance with similar studies [6,27]. Further, the median OS in the best prognostic group was approximately 3 years (39 months OUH vs. 33 months at KU/StO), which is an important reminder to clinicians that some patients may live relatively long even with a diagnosis of BM [28].

However, when evaluating individual prognostic variables, some differences between the hospitals became apparent. Most notably, the two Norwegian hospitals had a higher proportion of patients ≥ 70 years compared to KU. This is probably explained by the Gamma Knife surgery at KU [29,30,31] that potentially results in the selection of older patients to this therapy when referred for neurosurgical intervention. Similarly, a higher percentage of patients with breast cancer at KU could be due to a difference in referral practice combined with a higher incidence of breast cancer in Sweden [32]. The difference in the percentage of patients with poor ECOG performance status (>2) between the hospitals could be due to interrater variability, a problem previously described in cancer populations [33]. The difference could also be due to local discrepancies regarding the significance of performance status and selection for BM surgery.

### 4.2. Validity of the Preoperative Prognostic Index

The preoperative prognostic index successfully identified patients with longer OS in the validation groups at KU and StO but did not differentiate between the two groups with the poorest prognosis (Groups 2 and 3). This could partly be due to a lower number of patients in the two smallest groups. Based on these findings, we chose to adjust the preoperative prognostic index by merging the two poorest groups, creating three prognostic categories instead of four to accommodate external validity. Further, having three instead of four groups simplifies the use and interpretation in a clinical setting. We compared all hazard ratios from the validation data set with the derivation data set and all were comparable, with exceptions for ECOG status > 2 and the presence of breast cancer, which had less extreme values in the validation data set. This may explain the slightly lower concordance in the validation set.

### 4.3. Clinical Importance

A decision about surgical resection is usually done with careful consideration by a multidisciplinary tumor board. Even though some patients receive resection due to acute hydrocephalus (14% had hydrocephalus in our derivation data set [3]) or the need for a histopathological diagnosis, the aim is often therapeutic and based on the prognostic outlook of the individual patient. While stereotactic radiotherapy is a good treatment option, lesions may be too large, and neurological deficits may improve more promptly with surgical resection [21]. Whole brain radiation therapy is also an option, but in patients with longer expected survival time, or with single BM, this modality may not be recommended due to known side effects [34,35,36]. Traditionally, the focus of prognostic tools has been to identify patients with poor survival and to reduce the risk of potential overtreatment. The RPA and GPA prognostic tools were useful in this regard. However, they are based on outdated clinical data, due to the rapid development of new cancer therapies in the last three decades. Given the advances in the treatment of metastatic cancers, predicting longevity is also important in treatment selection. In the best prognostic groups, avoiding long-term side effects of treatment is important and should be considered in clinical decision-making. For many of these patients, surgical resection may be the best alternative even if there is not a need for acute decompression and treatment of hydrocephalus. In addition, in cases of histopathological uncertainty, estimating the prognosis of the patients may aid the decision between performing a simple biopsy or a gross total resection of a BM.

Our adjusted preoperative prognostic index is solely based on patients who underwent surgery for BM and allows for uncertainty regarding histopathology and extracranial disease. In contrast, well-established prognostic scores are based on patients who received a variety of treatments. In addition, the newly updated ds-GPA does not allow for unknown primary cancer. When it comes to surgery for BM, one-third of the patients may present with BM as the first sign of disease [3,26,37]. Our index does not contain molecular data such as EGFR mutation status in lung cancer patients, HER2 status in breast cancer patients and BRAF mutation status in melanoma patients. For a high fraction of patients such molecular data is unknown at the time of evaluation for craniotomy. In addition, while the above mutations are associated with survival, the association is uncertain in patients with BM [38,39], and more studies are warranted to establish the role of oncogenic mutations in these patients. Further, studies have also shown discrepancies between mutation status in primary tumors and their BM [40,41].

We believe that a prognostic index based on patients who underwent surgery for BM in the last decade is useful for patients and clinicians during joint decision-making. The index should be further validated in a prospective cohort of patients, preferably in patient populations outside of the Nordic countries, to increase validity. To ease the clinical use of the index, we further hope to create a readily accessible online calculator based on the index. Future studies should also include patient-reported outcome measures to investigate patient perspectives and symptoms in patients with short and long OS after treatment.

### 4.4. Strengths and Limitations

The study describes a high number of patients treated with BM surgery within geographically defined regions. We have used data from three treatment centers. The centers have different referral practices and as such somewhat different patient populations. In addition, there is significant heterogeneity within the populations on the variables selected for scoring prognosis. Study limitations include the retrospective collection of data and that the analysis was performed on patients who were already selected for surgery for BM. Further data on patients who were evaluated and declined for surgery would likely improve the index but are not available.

## 5. Conclusions

We believe current prognostic tools are insufficient for patients considered for surgery for BM since they either are based on very old data or require histopathological information that may be unavailable preoperatively. To facilitate decision-making for patients and clinicians, we created a preoperative prognostic index for these patients. The index categorizes patients into three prognostic groups with different expected survival times. We evaluated the index on external cohorts of patients, thereby demonstrating its validity and usefulness. The preoperative prognostic index can provide information on prognosis for clinicians and patients in the shared process of selecting the right treatment for the right patient with BM (Appendix A).

## Figures and Tables

**Figure 1 cancers-15-03174-f001:**
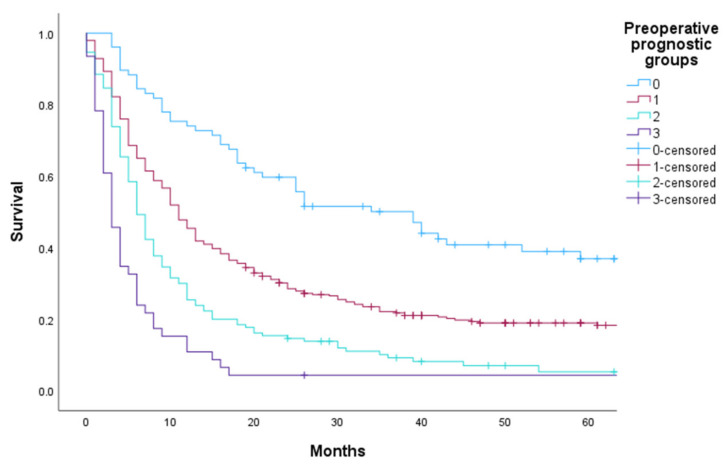
OS at OUH by preoperative prognostic index groups.

**Figure 2 cancers-15-03174-f002:**
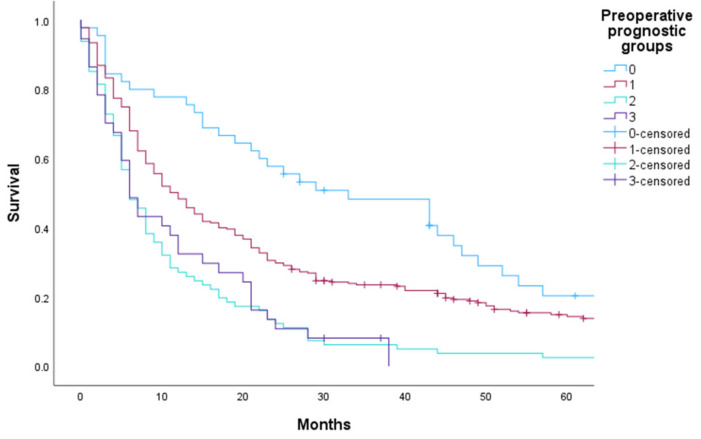
OS at KU/StO by preoperative prognostic index groups.

**Figure 3 cancers-15-03174-f003:**
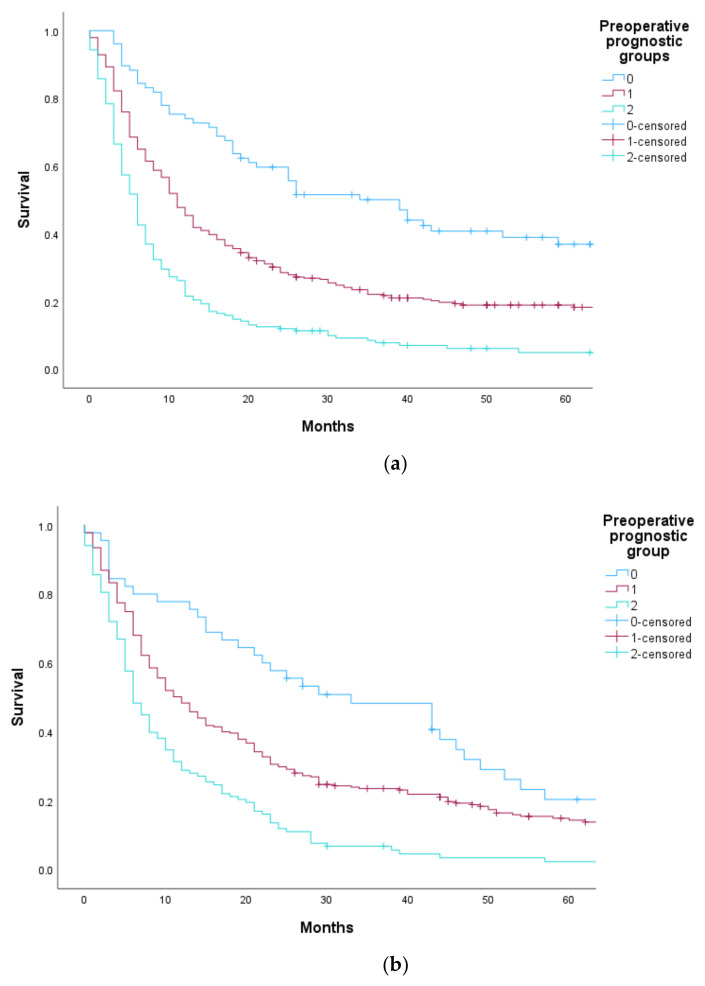
(**a**) OS at OUH by adjusted preoperative prognostic index groups. (**b**) OS at KU/StO by adjusted preoperative prognostic groups.

**Table 1 cancers-15-03174-t001:** Cox regression analysis as the basis for the preoperative prognostic index. We summarized **B** and obtained the preoperative prognostic index score.

Prognostic Variables ^1^	B	*p*-Value	HR (95% CI)
Age ≥ 70?	0.514	<0.001	1.67 (1.36–2.05)
ECOG > 2?	0.489	<0.001	1.63 (1.31–2.04)
Breast cancer?	−0.710	<0.001	0.49 (0.33–0.73)
Colorectal cancer?	0.350	0.036	1.42 (1.02–197)
Progressive extracranial disease ^2^ or synchronous brain metastases ^3^	0.345	<0.001	1.41 (1.17–1.71)
Extracranial metastases present? ^4^	0.394	<0.001	1.48 (1.23–1.79)
Previous chemotherapy?	0.295	0.009	1.34 (1.08–1.68)
Number of BM > 4?	0.439	0.010	1.55 (1.11–2.17)
Prognostic Index Score			Group
−1.0–0.0			0
>0.0–1.0			1
>1.0–1.4			2
1.4–2.9			3

^1^: Unknown = No. ^2^: Progressive extracranial disease: growing primary tumor or metastases, or new metastases three months prior to BM surgery. ^3^: Synchronous BM: primary tumor discovered within one month prior to BM surgery, or BM as first sign of disease. ^4^: Extracranial metastases were defined as documented extracranial metastases at any point prior to brain surgery, not including local lymph node infiltration.

**Table 2 cancers-15-03174-t002:** Preoperative prognostic index groups and OS at OUH *.

Prognostic Group	Prognostic Index Score	Total N (%)	Median OS in Months (95% CI)	HR (95% CI)
0	−1.0–0.0	77 (13)	39 (24–54)	1
1	>0.0–1.0	337 (57)	11 (10–12)	1.93 (1.42–2.62)
2	>1.0–1.4	130 (22)	6 (5–7)	3.1 (2.19–4.32)
3	>1.4–2.9	46 (8)	3 (2–4)	5.4 (3.56–8.19)
Overall		590 (100)	10 (9–11)	

* Harrell’s C = 0.62.

**Table 3 cancers-15-03174-t003:** General characteristics for each patient population.

	Oslo University Hospital (590)	Karolinska University Hospital (261)	St.Olavs University Hospital (177)
Age ≥ 70	148 (25%)	39 (15%) *	63 (36%)
ECOG > 2	111 (19%)	73 (28%) *	7 (4%) *
Breast cancer	53 (9%)	54 (21%) *	25 (14%) *
Colorectal cancer	50 (9%)	19 (7%)	23 (13%)
Progressive extracranial disease/synchronous BM	322 (55%)	112 (43%) *	98 (55%)
Extracranial metastases	312 (53%)	133 (51%)	80 (45%)
Previous chemotherapy	203 (34%)	155 (59%) *	98 (55%) *
Number of BM > 4	43 (7%)	13 (5%)	8 (5%)

* Statistically significant difference compared to OUH (*p* < 0.05).

**Table 4 cancers-15-03174-t004:** OS at KU/StO by preoperative prognostic index groups *.

Prognostic Group	Total N (%)	Median OS in Months (95% CI)	HR (95% CI)
0	45 (10)	33 (19–47)	
1	275 (63)	12 (9–15)	1.60 (1.12–2.29)
2	81 (18)	6 (4–8)	2.85 (1.90–4.28)
3	37 (8)	6 (4–8)	2.73 (1.70–4.40)
Overall	438 (100)	10 (8–12)	

* Harrell’s C = 0.58.

**Table 5 cancers-15-03174-t005:** OS at all hospitals by adjusted preoperative prognostic index groups.

Prognostic Group	Prognostic Index Score	Oslo University Hospital *			Karolinska/St.Olavs **		
		Total N (%)	Median OS in Months (95% CI)	HR (95% CI)	Total N (%)	Median OS in Months (95% CI)	HR (95% CI)
0	−1.0–0.0	77 (13)	39 (24–54)	1	45 (10)	33 (19–47)	1
1	>0.0–1.0	337 (57)	11 (10–12)	1.92 (1.41–2.62)	275 (63)	12 (9–14)	1.60 (1.12–2.29)
2	>1.0–2.9	176 (30)	6 (5–7)	3.46 (2.50–4.80)	118 (27)	6 (5–7)	2.81 (1.91–4.14)
Overall		590 (100)	10 (9–11)		438 (100)	10 (8–12)	

* Harrell’s C = 0.61, ** Harrell’s C = 0.58.

## Data Availability

The data used in this study are available on request from the corresponding author.

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
