# Peer review of "Preoperative Prognostic Index for Patients with Brain Metastases—A Population-Based Multi-Centre Study"

_cancers, 2023, doi:10.3390/cancers15123174_

Round 1
Reviewer 1 Report (New Reviewer)
Winther et al proposed a new preoperative prognostic index in brain metastasis surgery, utilizing a specific factors.
Simple summary/Abstract: Adequately summarizes the contents of the article. The abstract needs to state that this is meant to improve on recursive partitioning analysis (RPA), the current standard for prognostication in brain metastases.
Introduction: Provides an adequate background and the reason for the study, to improve on RPA which has long been used in the outpatient and clinical trial setting for brain metastases in general. The authors should indicate why new preoperative prognostic index in brain metastasis surgery when the RPA at the time of presentation has been adequate to determine prognosis and guide surgical decisions.
Materials/methods: The preoperative prognostic index was created based on a retrospective analysis of 590 patients who underwent surgery for brain metastases from 2011-2018 at Oslo University Hospital. The significant preoperative prognostic variables derived were age greater than or equal to 70, ECOG score greater than 2, breast cancer, colorectal cancer, progressive intracranial disease or synchronous brain metastasis, presence of extracranial metastases, previous chemotherapy and greater than 4 brain metastases. Validation was derived from retrospective reviews at 2 other institutions (Karolinka and St. Olav University). Data collection, statistical analysis and ethics approval were adequately described.
Results: Four prognostic groups (0-3) were consolidated to three based on similarities in group 2 and group 3. Groups were based on a prognostic index score range. How these were derived were not adequately documented in either materials/methods or results.
Discussion: Adequately analyzes the results. The authors need to address how decision making would be any different if RPA was used in decision making.
Conclusions: Adequately describes the findings in the discussion. Again, comparison with the much simpler RPA is needed.
Figures/tables: The footnotes in table 1a do not correspond to the citations in the table. Footnotes 2 and 3 should be consolidated and footnote 4 should be 3. Tables 3a and 3b should be reconsidered as these can be incorporated into the text.
Author Response
Please see the attachment

Reviewer 2 Report (New Reviewer)
The authors propose with this retrospective study an 8-point prognostic predictive score to identify the patient who best benefits from surgical treatment of brain metastases. This is an interesting study, considering the need on the part of the surgeon many times, to treat patients with unknown primary diagnosis. Therefore, I appreciate that all the various tumor species were considered as one group. However, I am intrigued by the choice of some variables put into the analysis: the presence of metastases from breast from colorectal and lung cancer should have notable differences from patients with metastases from melanoma considering in the last decade of biologic therapy with anti-BRAF-V600 drugs and the response they also have on metastases. In addition, metastases from melanoma when they debut after biologic therapy tend to be more de-differentiated and aggressive than others (Armocida D, Marzetti F, Pesce A, Caporlingua A, D'Angelo L, Santoro A. Purely Meningeal Intracranial Relapse of Melanoma Brain Metastases After Surgical Resection and Immunotherapy as a Unique Disease Progression Pattern: Our Experience and Review of the Literature. World Neurosurg. 2020 Feb;134:150-154. doi: 10.1016/j.wneu.2019.10.101. Epub 2019 Nov 18. PMID: 31751613.). Secondly, I am very surprised that the location of the metastasis even in numbers less than 4 was not taken into account for the score track, should a deep periventricular metastasis versus one in the frontal location have a different prognostic impact? Tumor volume? Is the type of surgery specified? Is the extent of resection specified?
I do not want to suggest changing the idea of the score, but I think these points should be considered in the discussion.
I suggest putting the suggested score in a more immediate graphic, and include some example pictures of a good and bad prognosis case.
Good general quality, minor spelling
Author Response
Please see the attachment

Reviewer 3 Report (New Reviewer)
This is a very relevant and important study, the authors built a brain metastases prediction model based on Cox-regression analyses.
There are some areas of opportunities to improve the manuscript:
1) Methods should describe all steps taken to build the model in order to replicate their process by others. For example, how did authors come with the values for each predictive marker? if the beta values were not used; how was the score built?, how many points were given to each predictive marker?
2) Numbers do not add up and the difference should be justified. For example, n from OUH was 590, KUH 263, and StOUH 184 (Table 3). KUH+StOUH = 705. Nevertheless, in Table 4 MOS was presented on information from only 442 patients, and in Table 5 the n from Karolinska/St.Olavs was 445.
3) From what the authors wrote, it reads as if the model was built in a training (derivation) dataset (OUH n 590) and validated in a cohort of 705 patients (KUH+StOUH). Authors only present the Cox-prediction model for the prognostic groups (Table 2, 3, and 4). Were the beta and HR of the predictive markers the same between the training and validation datasets?
4) Once a prediction model is created, at least a Brier test, AUC or Akaike information criteria should be measured and presented.
5) When creating a prediction model, it is ideal to follow guidelines; i.e. TRIPOD guidelines
6) What was the value of each predictive marker used to enter into the model? For example, having number of BM >4 was considered as 1 point? 0.5 points? -1 point?
7) Were all variables from Table 1a used? ibidem 6
8) Figures should have MOS (95% CI) log rank P values or HR (95% CI) and P values. Ideally survival tables should be added.
9) Was this model generated in a specific population/race? If so, the generalization could profit from including other populations (this could be added to the discussion).
10) Did patients receive postoperative radiotherapy? Wwre ther surgical complications?
11)
English writing is good and can easily be read.
Round 2
Reviewer 1 Report (New Reviewer)
All prior concerns were adequately addressed. No further comments.
Author Response
Thank you
Reviewer 2 Report (New Reviewer)
The authors have made a timely and accurate review making the paper more usable and understandable. However, some parts of the discussion here are too poorly addressed "The RPA and GPA prognostic tools were useful in this 271 regard. However, they are based on outdated clinical data, due to the rapid 272 development of new cancer therapies the last three decades. Given the advances in the 273 treatment of metastatic cancers, predicting longevity is also important in treatment 274 selection." how then is it that molecular variables are not considered in establishing prognosis? Recently it has been shown that EGFR expression also in BM from lung cancer is responsible for a certain type of manifestation and clinical course (See and cite: doi: 10.3390/jcm12103372) as well as BM from breast and the role with BRCA gene expression. The authors should mention the most recent studies or include them among the further studies section or in the limitations.
Author Response
Thank you again for your thorough review. We hope our new revision is satisfactory.
We agree that the role of oncogenic mutations in patients with brain metastases is important and complex and should be discussed more in-depth. We have added a paragraph with references in the Discussion section on page 9, lines 287-294.
This manuscript is a resubmission of an earlier submission. The following is a list of the peer review reports and author responses from that submission.
Round 1
Reviewer 1 Report
The authors have devised a prognostic index specifically for patients undergoing resection of brain metastases using a large cohort of operated patients in Norway, and validated it in a Swedish cohort. This is a very interesting work that will help neurosurgeons and oncologist determine the best patients who should undergo resection of brain metastases, versus other management modality. The study is well done. My only concern is the ease of the index, as the B value is not immediately intuitive and easy to remember. But this could be easily corrected by a smartphone or web app. The authors are to be congratulated for this work.
Author Response
Thank you for the comments. We agree that an online calculator would be a feasible method to ease the clinical use of the index. We have added a remark on this topic on page 9, line 249. With the short time available for review, we have not been able to set this up, but will do so shortly.
Reviewer 2 Report
I thank the Authors for the interesting paper exploring a new prognostic score for patients underwent surgery for brain mts.
The score was test in 1 center and validated in other 2.
The main limit is the retrospective nature however the Authors well underline d this limitation.
We already have a prognostic score for brain mts, I would recommend to develop more in deep a comparison between the present score and the well validated in literature in the Discussion session.
Author Response
We thank you for your review and comments. We understand your concerns and agree with you regarding the need for a deeper discussion of the difference between our new prognostic index and current prognostication scores for brain metastases. There are several differences, such as the possibility of unknown primary cancer. We have added this to the Introduction section on page 2, lines 70-81, and in the Discussion section page 9, lines 241-245.
Reviewer 3 Report
We have several prognostic indices for brain metastases (BM) patients. Authors claim that a separate prognostic index for patients referred for surgery of BM is useful and for that reason they performed an extensive work to select prognostic factors from their retrospective cohort of surgically treated patients. They further validated their findings on the independent data set from other centers.
My main objection for this study is the low utility of the findings from this study. Prognostic factors for BM are well known; the biggest one being performance status, then the presence of extracranial metastases. The indications for craniotomy are well defined, with the biggest one an emergency of waiving an intracranial hypertension, but also other, like a size of the tumor, a need for histo-pathological or molecular diagnosis. I don't see how the proposed index may serve in such indications, if treatment is necessary for emergency or a therapeutical decision on systemic treatment.
Additionally, in many cases, when immediate surgical intervention is not necessary, better tolerated treatment, like radio-surgery or fractionated stereotactic RT are proposed. This problem is not discussed by the authors.
Author Response
Thank you for your extensive and insightful comments regarding our manuscript. We appreciate the comment on the utility of the findings, as this is the most important point for us to address in the manuscript. Thus, we have made some efforts to elaborate on this. We do not find that the indications for craniotomy are well defined. The majority of patients undergoing surgery for brain metastases do not have acute hydrocephalus, and the decision to operate is often therapeutic and based on many factors, such as location and size of the tumour, comorbidities, extent of disease, performance status, previous treatment and age of the patient. International guidelines for surgery are general and the multidisciplinary team has to weigh all factors before deciding. We believe that a prognostic index created for this exact purpose may be useful. This is particularly so because all relevant information one has at hand, may facilitate the final decision about surgery or not. We revised the manuscript to addressed this more in-depth in the Discussion section on pages 8-9, lines 223-245.
We agree that stereotactic radiotherapy is a well-tolerated management option. However, in many cases, radiotherapy is not advisable, for instance due to the size of the brain metastasis. We have addressed this further in the discussion section on page 8, lines 223-240.
Round 2
Reviewer 2 Report
I thank you the Authors for the improvements, however I think that the paper does not change and add sufficient knowledge to the specific issue